Visual complexity modelling based on image features fusion of multiple kernels

Fernandez-Lozano Carlos 1
Carballal Adrian adrian.carballal@udc.es 1
Machado Penousal 2
Santos Antonino 1
Romero Juan 1
1 Computer Science Department, Faculty of Computer Science, University of A Coruña , A Coruña , Spain
2 CISUC, Department of Informatics Engineering, University of Coimbra , Coimbra , Portugal
Peng Yonghong
Electronic publication date: 2019 Jul 18
Publication date: 2019
Volume: 7
Electronic Location ID: e7075
Received 2018 Oct 16; Accepted 2019 May 4
Copyright: ©2019 Fernandez-Lozano et al.
Copyright year: 2019
Copyright holder: Fernandez-Lozano et al.
License: This is an open access article distributed under the terms of the Creative Commons Attribution License, which permits unrestricted use, distribution, reproduction and adaptation in any medium and for any purpose provided that it is properly attributed. For attribution, the original author(s), title, publication source (PeerJ) and either DOI or URL of the article must be cited.
License URL: https://creativecommons.org/licenses/by/4.0/

Keywords: Correlation, Machine learning, Zipf’s law, Compression error, Visual stimuli, Visual complexity

Funding: General Directorate of Culture, Education and University Management of Xunta de Galicia GRC2014/049 The European Fund for Regional Development (FEDER) allocated by the European Union The Portuguese Foundation for Science and Technology for the development of project SBIRC PTDC/EIA–EIA/115667/2009 Xunta de Galicia XUGA-PGIDIT-10TIC105008-PR Spanish Ministry for Science and Technology TIN2008-06562/TIN The Juan de la Cierva fellowship program by the Spanish Ministry of Economy and Competitiveness FJCI-2015-26071 This work is supported by the General Directorate of Culture, Education and University Management of Xunta de Galicia (Ref. GRC2014/049) and the European Fund for Regional Development (FEDER) allocated by the European Union, the Portuguese Foundation for Science and Technology for the development of project SBIRC (Ref. PTDC/EIA–EIA/115667/2009), Xunta de Galicia (Ref. XUGA-PGIDIT-10TIC105008-PR) and the Spanish Ministry for Science and Technology (Ref. TIN2008-06562/TIN) and the Juan de la Cierva fellowship program by the Spanish Ministry of Economy and Competitiveness (Carlos Fernandez-Lozano, Ref. FJCI-2015-26071). NVIDIA Corporation donated the Titan Xp GPU used for this research. The funders had no role in study design, data collection and analysis, decision to publish, or preparation of the manuscript.

==============================
Humans’ perception of visual complexity is often regarded as one of the key principles of aesthetic order, and is intimately related to the physiological, neurological and, possibly, psychological characteristics of the human mind. For these reasons, creating accurate computational models of visual complexity is a demanding task. Building upon on previous work in the field (Forsythe et al., 2011; Machado et al., 2015) we explore the use of Machine Learning techniques to create computational models of visual complexity. For that purpose, we use a dataset composed of 800 visual stimuli divided into five categories, describing each stimulus by 329 features based on edge detection, compression error and Zipf’s law. In an initial stage, a comparative analysis of representative state-of-the-art Machine Learning approaches is performed. Subsequently, we conduct an exhaustive outlier analysis. We analyze the impact of removing the extreme outliers, concluding that Feature Selection Multiple Kernel Learning obtains the best results, yielding an average correlation to humans’ perception of complexity of 0.71 with only twenty-two features. These results outperform the current state-of-the-art, showing the potential of this technique for regression.

Introduction

When looking at an image one has an immediate sense of its complexity. This feeling is related to the physiological, neurological and, possibly, psychological characteristics of the human mind, namely with the way the human brain processes images (Zeki, 1999). Although it has a neuro-physiological basis, the perception of visual complexity depends on many factors. For instance, during infancy the exposure to visual stimuli is vital for defining the main pathways of the visual cortex; likewise, our familiarity with given stimuli may affect the way we react to it. As such, although we all have a sense of visual complexity, a measure of complexity is inherently subjective, varying from one viewer to the other, and even depending on several circumstantial factors (e.g., fatigue). Therefore, creating accurate computational models of visual complexity is a demanding task, since the creation of an accurate model may ultimately require modeling the visual cortex and even life experiences.

Humans’ perception of visual complexity is often regarded as one of the key principles of aesthetic order. Additionally, starting with (Birkhoff, 1933), there is a high number of psychological papers that propose relations between visual complexity and aesthetic value. In the state-of-the-art some of them will be discussed. Visual complexity has also been widely employed in Human–Computer interaction studies (Tuch et al., 2012; Tuch et al., 2009; Smeddinck, Gerling & Tiemkeo, 2013; Stickel, Ebner & Holzinger, 2010), GUIs design (Miniukovich, Sulpizio & De Angeli, 2018; Miniukovich & De Angeli, 2014), studies on user attention (Harper, Michailidou & Stevens, 2009; Michailidou, 2008), and even related to user trust (Tseng & Tseng, 2014).

In this paper, we build upon the work of Forsythe et al. (2011) and Machado et al. (2015) in this area. In 2011, Forsythe employs a dataset created by Cela-Conde et al. (2009) to measure the correlations between visual complexity, as perceived by humans, and several computer generated measures. In 2015, this dataset was used by Machado et al. (2015), who employed a novel set of computer generated metrics and Artificial Neural Netwoks (ANNs) as a Machine Learning technique, in order to predict the visual complexity of the dataset images.

The dataset and the metrics are the same as those used by Machado et al. (2015). We expand the work by exploring different Machine Learning techniques and conducting an in-depth analysis of the results. In particular, we refer to the outlier analysis, which allows us to determine the stimuli that are more problematic for the prediction task.

An initial comparison of the results obtained by the tested Machine Learning (ML) techniques, allows us to conclude that the best approach is ENET, yielding results that surpass the current state-of-the-art, obtaining a R-Squared correlation value to humans’ perceived complexity of 0.71, while the previous best was 0.69 (Machado et al., 2015).

The outlier analysis allowed us to identify problematic stimuli, which are likely to be inappropriate to training and testing purposes, seriously biasing the results. Once these stimuli are removed and the models retrained, the best ML approach becomes FSMKL, yielding a R-Squared correlation value to humans’ perceived complexity of 0.71, which surpasses the best result obtained by Machado et al. (2015).

Feature Selection Multiple Kernel Learning allows identifying a small subset (22 out of 329), resulting in a good prediction of perceived complexity. These results reinforce previous findings, allowing us to conclude that the Value and Saturation channels of the images are the most informative ones, while the Hue channel seems to be the least useful for complexity estimation. Furthermore, Canny edge filter, JPEG and Fractal compression and Zipf’s law metrics are the most relevant in this study.

The following section will describe briefly the state-of-the-art about measuring complexity and some psychological papers that propose the relationship between image complexity and aesthetics.

State-of-the-art

The first methods for measuring visual complexity take into account the number of elements (lines, angles or polygons) and the regularity, irregularity, and heterogeneity of those elements (Birkhoff, 1933; Eysenck & Castle, 1971). The stimuli employed were typically created by the researchers with a different set of polygons, in order to allow the manual counting of elements. Using this approach, in 1933, Birkhoff (1933) formulated complexity as M = O/C, where “M” is the aesthetic measure or value, “O” is the aesthetic order, and “C” is complexity. In other words, beauty increases as complexity decreases. Thereafter, Eysenck & Castle (1971) studied the correlation between Birkhoff’s measure and complexity measuring fifty figures against their seven-point-scale aesthetic pleasantness judgments from 1,100 participants (100 artists and 1,000 non-artists). Only very slight differences were observed between the experimental and control groups according to these authors.

More recent studies on experimental aesthetics focus on how the exposure time (Schwabe et al., 2018) affects processing artistic images, examine what intrinsic and extrinsic factors affect the aesthetic response to images (Mullin et al., 2017), and how “good composition” or “visual rightness” are revealed according to basic features related to edge orientation and luminance (Redies, Brachmann & Wagemans, 2017). Other similar features have also been recently used in image retrieval research (Ali et al., 2016a; Ali et al., 2018; Zafar et al., 2018a).

In 2006, Lempel & Ziv (2006) developed an algorithm to measure visual complexity. Their algorithm was based on the smallest computer program required to store/produce an image as a basis for the compression techniques we use today. The idea is based on the theory that the minimum length of the code required to describe an image is an adequate measure of complexity (Leeuwenberg, 1969).

Inspired by this concept, Donderi (2006) argued that the adequacy of image compression techniques to predict subjective complexity was directly related to the Algorithmic Information Theory. According to Aksentijevic and Gibson, the “algorithmic complexity is defined in terms of the length of the shortest algorithm in any programming language, which computes a particular binary string” (Aksentijevic & Gibson, 2012). Aiming to address this approach, various edge detection methods such as Perimeter Detection, Canny, and others, based on phase congruency proved to be a reliable way of measuring complexity in the visual domain (Forsythe et al., 2011; Marin & Leder, 2013).

The most popular and widely used method to determine visual complexity is deriving a set of images and asking some participants to rate their complexity (Cycowicz et al., 1997; Alario & Ferrand, 1999). Following this methodological line, Forsythe et al. (2011) examined the performance of a series of metrics related to JPEG 2000, GIF compression and perimeter detection of over 800 visual stimuli evaluated by 240 humans who provided ratings with a bound and previously indicated range of complexity (Cela-Conde et al., 2009). According to the authors, GIF compression was correlated most strongly with human judgments of complexity for 800 artistic and nonartistic, abstract and representational images. Their results showed that this computational measure was significantly correlated with judged complexity getting a R-Squared = 0.5476.

In this context, Marin & Leder (2013) compared several computational measures correlated with participants’ complexity ratings of different kinds of materials. They found that TIFF file size (R-Squared = 0.2809) and JPEG file size (R-Squared = 0.2704) correlated strongest with subjective complexity ratings rather than measures of perimeter detection using a subset of stimuli selected from the International Affective Picture System. The differences between the results obtained by Forsythe et al. and Marin and Leder may have to do with both materials and procedure.

Recently, Machado et al. (2015) have proposed a wide range of new possible complexity estimates. These features were based on image compression error and Zipf’s law (Zipf, 1949). Every feature was calculated for each image by applying different edge detection filters on all color channels. For edge detection, authors examined the performance of the well-known filters Sobel (Sobel, 1990) and Canny (Canny, 1986). Consequently, a total of 329 features based on seven metrics applied to three color channels (Hue, Saturation and Value), and using both above-mentioned filters, were extracted (Guyon et al., 2006).

According to Machado et al. (2015), estimates which share similarities with the perimeter detection method employed by Forsythe et al. (2011), which directly measure the percentage of the pixels of the image that correspond to edges, obtain better results than those from the state-of-the-art. Otherwise, similar metrics to those obtained by Forsythe et al. (2011) using GIF compression, based on JPEG and Fractal compression error and no edge detection application, obtain similar results. Nevertheless, features directly related to the measurement of the number of edges are a better estimate of image complexity (R-Squared = 0.5806) than the perimeter detection method employed by Forsythe et al. (2011). Finally, the authors stated that the highest overall correlations were obtained using JPEG compression, after previously applying Canny (R-Squared = 0.5868) and Sobel (R-Squared = 0.5944) edge detector filters.

Machado et al. (2015) noted that edge density and compression error were the best predictors of participants’ complexity ratings, suggesting that the perceptual and cognitive processes involved in detecting edges and dealing with non-redundant information play crucial roles in the subjective experience of complexity. Furthermore, analyzing the correlation of individual estimate, they tried to evaluate the results by combining their feature vector and Artificial Neural Networks (ANN). Their aim was to improve the results using a computational method to predict human’s complexity scores. They reported an increase of R-Squared = 0.6939 using all proposed features as inputs and an MLP. Despite the fact that this result upturned the correlation regarding a unique metric, it should be noted that the difference entailed the use of 329 features instead of a single one. They tested with different combinations of inputs, concluding that the one that performed best was the one that had access to all the proposed features.

Considering such a proposal as an initial approach, the choice of a model based on ANN is not always the best choice in tasks exclusively related to prediction. Neural networks in general and Multilayer Perceptrons in particular tend to present overfitting, especially with few samples (Lawrence, Giles & Tsoi, 1997). There are methods to minimize such problems, such as the use of the dropout technique or combining the predictors of many different models (Hinton et al., 2012) but none of them was applied. Therefore, in this study, other models based on Machine Learning from the latest state-of-the-art applied to the same input data used by Machado et al. (2015) are proposed. The objective is not only to improve existing results, but also to conduct a statistically more rigorous study, which may confirm the presence of outliers and their relevance in the process of regression.

Materials and Methods

The authors tested the different computational models using 10-fold cross-validation to split the data and 50 runs per model in order to evaluate the performance across different experiments and study the standard deviation. The performance of the models was evaluated using R-squared (R2) and Root Mean Squared Error (RMSE) as well as the number of features (metrics).

Stimulus selection

In the field of aesthetic psychology there are numerous works in which human subjects were used to evaluate different stimuli following aesthetic criteria (Forsythe et al., 2011; Martinez & Benavente, 1998; Amirshahi et al., 2014; Corchs et al., 2016; Marin & Leder, 2016; Melmer et al., 2013; Koch, Denzler & Redies, 2010; Georghiades, Belhumeur & Kriegman, 2001; Van Hateren & van der Schaaf, 1998; Redies, Hasenstein & Denzler, 2007; Redies et al., 2012). For example, (Schettino et al., 2016) recruited seventeen male volunteers (age range from 19 to 33) to evaluate one-hundred pictures depicting various everyday scenes (e.g., people at the supermarket, at the restaurant, playing music, or doing sport), as well as nude female bodies and heterosexual interactions, were selected from the IAPS (Lang, Bradley & Cuthbert, 2008).

In (Street et al., 2016), four hundred and forty-three participants, 228 men and 204 women, aged between 17 and 88 evaluate 81 abstract monochrome fractal images (9 full sets of 9 iterations of FD) generated using the mid-point displacement technique.

Besides, (Lyssenko, Redies & Hayn-Leichsenring, 2016), 19 participants (19–37 years old) chose 79 images from a collection of 150 images of abstract artworks that was compiled by Hayn-Leichsenring, Lehmann & Redies (2017). Finally, Friedenberg & Liby (2016) selected twenty-five undergraduates (5 males and 20 female) from Manhattan College in New York to evaluate 10 images determined by the authors.

An overview of all reviewed datasets is shown in Fig. 1, making reference to the number and gender of the participants and in Fig. 2 where the available stimuli used in experimentation are listed. Taking both into account, (Forsythe et al., 2011) seems to be adequate for this research in terms of participants and stimuli used. The selection of the dataset obviously depended also on the availability of data (besides the large N of stimuli etc.). We only have access to the dataset used by Forsythe et al. (2011).

Figure 1 Number and gender of the participants in analyzed works of the state-of-the-art.

Figure 2 Available images comparison (A) and datasets of stimuli (B) in the analyzed works of the state-of-the-art.

Forsythe’s et al. stimuli

Stimuli were initially provided by Cela-Conde et al. (2009), containing a set of over 1500 digitalized images including abstract and representational images and artworks. The difference between representational and abstract stimuli relies on the presence or absence of explicit content. Artistic stimuli included reproductions of renowned artists’ paintings, all catalogued and exhibited in museums. The authors took 19th and 20th century paintings of different styles: realism, cubism, impressionism, and postimpressionism. Non-artistic stimuli were gathered from different book series and collections such as Boring Postcards (Parr, 1999; Parr, 2000) and CDs Master Clips Premium Image Collection (IMSI, San Rafael, CA). This category included artefacts, landscapes, urban scenes, and others considered of interest for their exhibition in museums. Artistic and non-artistic categories were defined analogously to the distinction method for popular art versus high art used by Winston & Cupchik (1992). According to Cela-Conde et al. ‘popular art emphasizes subject matter, especially its pleasing aspects, high art relies on a broader range of knowledge and emotions’.

Aiming to avoid the impact of familiarity, some images were either discarded or modified. Images containing clear views of human figures or faces, or portrayed emotional scenes were also eliminated. Stimuli with a mean distribution of pixels concentrated in both extremes of the histogram were discarded. All these modifications were focused on minimizing the influence of strange variables.

Additionally, all stimuli were set to 150 ppi, their size to 9 by 12 cm, the color spectrum was adjusted in all images to reduce the influence of psychophysical variables and the luminance of the stimuli was adjusted to between 370 and 390 lx. In some cases, author’s signature was removed manually for proper anonymization.

The final standardized set included 800 images grouped into 5 categories: abstract artistic (AA), abstract non-artistic (AN), representational artistic (RA), representational non-artistic (RN), and photographs of natural and human-made scenes (NHS).

Participants

According to Forsythe et al. (2011), two hundred and forty participants (112 men and 128 women) from the University of the Balearic Islands, without formal artistic training, took part in the study.

The 800 images were divided into 8 equally distributed sets of 100 images. At the beginning of the experiment, each participant was given an image with very low complexity (an icon) and other with high complexity (a city) as examples (Cela-Conde et al., 2009). Participants were given a definition of complexity as “the amount of detail or intricacy” according to the definition made by Snodgrass & Vanderwart (1980).

Then, participants were seated between 2 and 7 m from a visual display where 100 images were presented with ratio 16:9 and size 400 × 225 cm. Each image was displayed for 5 s. All participants rated these pictures on a scale from 1 to 5, 5 being categorized as very complex and 1 as very simple.

Figure 3 shows the scatter plot of the assessments made by men and women. This information is shown in order to explain the high degree of subjectivity of the task. In this case, the existing R-squared correlation is 0.8485. For the authors, this value is a good estimate of the maximum level achievable in this particular problem.

Computational models

The authors performed several experiments in order to select the best model using the R package (R Core Team, 2016) and MATLAB®. Some of the used computational models looked for the smallest subset of variables of the original set which provide a better performance (Blum & Langley, 1997), or at least equal to that obtained when using all the possible variables, considering this is a Feature Selection (FS) approach (Fernandez-Lozano et al., 2015; Saeys, Inza & Larrañaga, 2007; Bolón-Canedo, Sánchez-Maroño & Alonso-Betanzos, 2013; Jain & Zongker, 1997). There are mainly three different approaches for FS known as filter (Hall & Smith, 1999), wrapper (Kohavi & John, 1997) and embedded.

Figure 3 Complexity correlation between men and women’s rating of stimuli (Source: Forsythe et al. (2011)).

More specifically, the used methods are as follows: a recently proposed Feature Selection Multiple Kernel Learning (FSMKL) (Fernandez-Lozano et al., 2016b) which is a filter approach for classification tasks (Dash & Liu, 1997), the well-known Support Vector Machines—Recursive Feature Elimination (SVM-RFE) (Guyon et al., 2002; Chang & Lin, 2011; Maldonado, Weber & Basak, 2011), Elastic Net (ENET) by (Zou & Hastie, 2005), Lasso by (Tibshirani, 1996) which includes embedded approaches, Generalized Linear Model with Stepwise Feature Selection (GLM) by (Hocking, 1976) which selects features that minimizes the AIC score and the most basic standard Multiple Linear Regression (LM) without FS.

The capabilities of the RRegrs Package (Tsiliki et al., 2015) were enhanced in order to implement the SVM-RFE, ENET, Lasso, GLM and LM and to avoid finding the best model according to the proposed methodology since, according to (García et al., 2010) it should be done based on a null hypothesis test. This package was also enhanced in order to avoid the initial splitting process, and an external cross-validation process was performed to avoid selection bias as suggested by (Ambroise & McLachlan, 2002), and the last step was modified in order to easily extract the results for all the models. The FSMKL was also improved following the criteria in (Rakotomamonjy et al., 2008) in order to solve regression problems (Menden et al., 2017). Next, the initial proposed ranking criterion, proposed by Guyon et al. (2002) for SVM-RFE, was enhanced using w2 to measure the importance of each feature.

Finally, some other R packages were used in different parts of this study, more specifically: Caret (Max Kuhn et al., 2016), Kernlab (Karatzoglou et al., 2004), cvTools (Alfons, 2012), doMC (Analytics & Weston, 2015), car (Fox & Weisberg, 2011) and ggplot2 (Wickham, 2009).

In order to train the computational machine learning models, a novel methodology for the development of experimental designs was applied in regression problems with multiple machine learning regression algorithms (Fernandez-Lozano et al., 2016a).

Deep Neural Networks have been previously used in image and visual studies for handwritten digit recognition (Lecun et al., 1998), object recognition (Krizhevsky, Sutskever & Hinton, 2012) or image classification (Deng et al., 2009). Regarding to aesthetics, the most noteworthy contribution was the study conducted by Tan et al. (2016), were the researchers extracted 56 features normalized to [0, 1] and trained a novel aesthetics classifier based on an improved artificial neural network combined with an Autoencoder technique with photographs of high and low ratings to test the quality of photos for classification only. The experiments in the early stages of this study, along with other learning methods, showed no remarkable results, so they were not used in later stages.

Machine learning is widely used in several completely different fields; detection of regions of interests, image editing, texture analysis, visual aesthetic and quality assessment (Datta et al., 2006; Marchesotti et al., 2011; Romero et al., 2012; Fernandez-Lozano et al., 2015; Mata et al., 2018) and more recently in Carballal et al. (2019b) or for microbiome analysis (Liu et al., 2017; Roguet et al., 2018), authentication of tequilas (Pérez-Caballero et al., 2017; Andrade et al., 2017), pathogenic point mutations (Rogers et al., 2018) or forensic identification (Gómez et al., 2018). Finally, with regard to the extraction of characteristics from images, some recently published works have been revised (Xu, Wang & Wang, 2018; Ali et al., 2016b; Wang et al., 2018; Sun et al., 2018; Zafar et al., 2018b).

Results

The final set of experimental stimuli was composed of 800 images grouped into five categories: AA, AN, RA, RN and NHS. Six different Machine Learning computational models were used in order to evaluate visual complexity. Some of them were complex approaches that evaluated the dataset following a feature selection approach.

The models previously published in Machado et al. (2015) were used as a baseline for comparison purposes with our proposal. In this work, the authors identified as the best single feature for this particular problem, the one that calculated the JPEG compression error in the saturation color channel, with prior application of edge detection filters. With respect to the ANN used in this work, they were specific for four different configurations: NET1 contained all the metrics filters and extracted color channels, NET2 did not make the most of the edge detection filters, NET3 used the edge detection filters, but did not make the most of the proposed complexity estimates, whereas NET4 used only basic metrics (mean and standard deviation), without edge detection filters. These five state-of-the-art results are shown in Fig. 4 with lines of different colors.

The analysis started using six ML techniques for the visual complexity regression problem in order to compare the results with those previously obtained by Machado et al. (2015). In this paper, using all color channels, the authors reported four Artificial Neural Networks (ANN) with R2 values ranging from 0.6938 to 0.2218. As shown in Fig. 4 SVM-RFE and ENET outperformed the best published results, FSMKL achieved results very similar to the previously published ones, whilst LM, GLM and Lasso obtained poor results. The time that each technique needed to perform the 50 experiments is shown in Fig. 5; we avoided to specify the time for the GLM model as it needed more than 150 h. The best results for R-squared (median of the 50 experiments) were achieved with alpha = 0.8 for ENET (0.71), GLM (0.33) implemented internally Stepwise Feature Selection (AIC criterion), Lasso (0.52) with fraction = 0.1, SVMR-RFE (0.69) with bestsubsetsize = 256, sigma = 2−6, C = 10 and epsilon = 0.01, LM (0.26) and FSMKL (0.68).

Figure 4 R-squared comparison between the state-of-the-art results and the six computational methods used in this work.

The comparison attends to the application of (A) compression methods, (B) edge detection filters and (C) color channels.

Figure 5 Time (in hours) for each model.

From the best three models, the most stable one in terms of R-squared, RMSE and number of features was the novel FSMKL as shown in Fig. 6. An MKL approach was aimed at simultaneously learning a kernel and the associated parameters. In the current study, classical kernels were used: Gaussian (with sigma values of 0.1, 0.2, 0.3, 0.4, 0.5, 1 and 2) and polynomial (degrees 1, 2, 3 and 4). Thus, the importance (weight) of each kernel can be measured in the final solution. Furthermore, as the use of FSMKL was proposed,firstly all the features were ranked following a filter approach and thus, our searching space included for each group of features (with size n), one kernel per each size value of features (ranging from 2 to n). This means that the same feature could be available in different kernels with different weight value, and the particular sum of weights could be calculated for each feature in the final solution. Our proposal was initially based on that of SimpleMKL, where MKL was solved by using a projected gradient method and the overall complexity is tied to the one of the single kernel SVM algorithm, for more information please refer to Rakotomamonjy et al. (2008).

Figure 6 (A) Number of features and (B) RMSE of the best three computational models used in this work: FSMKL (0.56), SVM-RFE (0.58) and ENET (0.53).

FSMKL used the same twenty-two features in forty-one out of the fifty experiments whilst SVM-RFE used 192 and ENET 25. Table 1 and Fig. 7 show the twenty-two features employed in FSMKL. These features are mostly coherent with the findings of Machado et al. (2015).

Table 1 Set of features used by FSMKL, organized by importance.

The identified features were: the position (POS), the accumulated weight and the numbers of kernels in which they were used. All the features were identified using the terminology proposed in Machado et al. (2015).

Pos	Feature	Weight	Kernels	Pos	Feature	Weight	Kernels	
1	Fractal(NoFilter(S),High)a	2.627	3	12	JPEG(Canny(S),High)a	0.747	4	
2	Fractal(NoFilter(V),High)	2.627	3	13	JPEG(Canny(S),Medium)	0.747	4	
3	Fractal(NoFilter(V),Medium)	2.216	2	14	JPEG(NoFilter(S),Medium)	0.483	2	
4	Rank(NoFilter(S),R2)a	2.122	4	15	Fractal(Canny(S),High)a	0.393	2	
5	Rank(NoFilter(S),M)	2.122	4	16	Size(Canny(S),M)a	0.358	4	
6	JPEG(NoFilter(S),High)a	1.335	5	17	Size(Canny(V),M)	0.358	4	
7	JPEG(NoFilter(V),High)	1.335	3	18	JPEG(Canny(S),Low)	0.326	4	
8	Size(NoFilter(S),M)	1.195	4	19	Size(NoFilter(V),M)	0.149	2	
9	Size(NoFilter(H+CS),R2)a	1.195	4	20	Size(NoFilter(V),R2)	0.149	2	
10	Fractal(Canny(S),Low)	0.883	3	21	Rank(Canny(S),R2)	0.128	4	
11	Fractal(Canny(S),Medium)	0.883	3	22	Rank(Canny(S),M)a	0.128	1	
Notes.

a Features previously identified by Machado et al. (2015) as the best individual features for solving the problem.

Figure 7 Prevalence of features relating to metric family (A), edge detection (B) and HSV color channel (C) of the twenty-two recurrent features according to the FSMKL method.

Regarding the color channel, according to Machado et al. (2015), the Saturation color channel was more informative than the Value channel, which, in turn, was more informative than the Hue channel. According to the data shown in Table 1 and Fig. 7, this tendency is clear with 68% of the metrics related to the Saturation channel, 27% related to Value channel and only one metric related to Hue channel. This can be explained due to the type of images employed in the dataset. Most of the images are designs and paintings where the change in color can be made by change in the Saturation and not in the Value channel, so Saturation can be more relevant than Value. In real-life images (photographs) the most relevant channel is usually that of Value. Another fact that can be analyzed in Table 1 is that most of the features related to the Value channel have an equivalent features related to the Saturation channel, such as the case of features 1 and 2, features 6 and 7, features 8 and 19, etc. Thus, the system employs the same feature applied to both channels.

Regarding filters, in Machado et al. (2015) seven possibilities were considered: No_filter, Canny_all, Canny_horizontal, Canny_vertical, Sobel_all, Sobel_horizontal, Sobel_vertical. Out of these seven options, only two were included in the twenty-two selected features: Canny_all (45% of the metrics), and No_filter (55% of the metrics). None of the Sobel metrics was taken into account, as neither vertical nor horizontal versions of Sobel and Canny edge filters are used. In Machado et al. (2015) the features obtained using filters provided better individual results, the high individual correlation being JPEG(No_Filter(S),High) with r = 0.55. But, as shown in Table 1, the features with more weight were those without filters.

Regarding the families of metrics, all the families employed in Machado et al. (2015) were used by the FSMKL except for the very simple ones, related to average and standard deviation of pixel values. In terms of compression metrics, as happened with the channel before, there were pairs of features where only the type of compression changed. This is the case for features: 1–6, 2–7, 11–13 and 10–18. There were also pairs of features that only change the compression level, as in the pairs: 2–3, 10–11, 12–13. Although there was a very high correlation between JPEG and Fractal compression based families of features (r = 0.984 as shown in Machado et al. (2015)), it seems that FSMKL used these minimal differences to optimize the objective function. Regarding the Zipf’s Law based metrics, there was a strong presence in the features chosen by FSMKL with 45% percentage. As before, there were some pairs of features, where the only change was the Zipf output, as in features 4–5, 19–20, 21–22.

Due to the high stability of FSMKL and to the low number of features, it was decided to analyze possible outliers using this method, since the aim of a feature selection process is to find the lowest number of features that performs equally or ideally better than the full set. Technically, FSMKL is a filter feature selection approach and SVM-RFE and ENET are embedded approaches (Saeys, Inza & Larrañaga, 2007).

Outlier removal

In our opinion, some images had extreme values that could have affected the regression training of the algorithms. After training the system, an outlier analysis (Cook’s distance, studentized residuals and high leverage points) was performed in order to detect the samples that could be considered extreme outliers, and thus change the fit of the model. Furthermore, a diagnosis of regression analysis was carried out using diagnostic residual scatterplots of the Pearson residuals against the fitted values. The main objective of the four graphical methods in Fig. 8 is to assess the adequacy of the regression model and to find the possible outliers that generate problems and a decrease in performance (R. Dennis Cook, 1997). A Tukey’s test was performed with the data from Fig. 8A with the null hypothesis that the model is additive, and a p-value of 0.611 was obtained, which is not lower than the significance level α = 0.05 and the null hypothesis could not be rejected. In order to identify the influence of a particular image on the regression coefficient, the partial regression plot shown in Fig. 8B was plotted. Furthermore, the normality of the data in Fig. 8C was checked and an outlier test was performed using the Bonferroni correction with the null hypothesis that there were outliers in our data, thus the null hypothesis with a value of 0.5988 could not be rejected. Figure 8D shows a bubble-plot combining the display of studentized residuals, hat-values, and Cook’s distance (represented by the size of the circles).

Figure 8 Diagnosis of the regression analysis: (A) Residuals plot of fitted vs Pearson residuals (B) Influential variables in an added-variable plot (C) Normality qq-plot and (D) Influence plot (studentized residuals by hat values, with the areas of the circles representing the observations proportional to Cook’s distances).

After these analyses, it was decided to remove six of the images as shown in Fig. 9, and the algorithms were trained again, without these outliers. An improvement was observed in the R-squared cross-validation score in this new set of other fifty experimental runs. Figure 10A shows a comparison between the best three models in Fig. 4 with and without outliers. Under the removed images in Figs. 9A–9F, the following was plotted:

• the influence of each particular image on the final model according to Cook’s distance measure

• the studentized residuals (residuals divided by their estimated standard deviation)

• the p-values of the outliers test (Bonferroni correction)

• the hat-values in order to identify the influential observations

Regarding the visual content of the outliers, there were several aspects to consider. It should be noted that five out of six detected outliers belonged to the NHS group. In this respect, it should be taken into account that these outliers were well distributed, as it can be observed in the AVG values in Figs. 9A–9F. In Machado et al. (2015), the authors concluded that this particular group of images were different in several ways with respect to the other four. The most obvious one was that this group was composed of photographs while the others were designs and paintings.

Figure 9 Six images were found, which were clearly outliers (A–F). An outlier analysis was performed, paying attention to: (G) the influence of each particular image on the final model according to Cook’s distance measure, (H) the studentized residuals (residuals divided by their estimated standard deviation), (I) the p-values of the outlier test (Bonferroni correction) and (J) the hat-values in order to identify influential observations.

Figure 10 Outliers plots: (A) boxplot with the results of the best three methods (ENET, FSMKL and SVM-RFE) with and without the outliers. *, Statistically significant difference with a p-value < 2.2 × 10−16 according to a pairwise Wilcoxon test; **, Statistically significant difference according to a non-parametric Friedman test with Iman and Davenport correction with a p-value < 1.92 × 10−41. At the bottom of the panel, a median-based contrast estimation heatmap. In (B) the residues are plotted by coloring each of them according to the category to which the image belongs.

Four out of the five outliers of the NHS group had clear gradients (outliers 2, 4, 5 and 6). The gradients in a photograph are very different from the gradients that can be found in a painting and nothing like those in a design. They are very simple for humans but they may be complex for algorithms regarding edges and error of compression. Moreover, the gradients in theses images were part of the background, so humans focus less on them. Outlier 4 was very peculiar, since it could look similar to a simple design (plain background with a simple form) but in fact the background was a gradient and the figure had a lot of noise. Outliers 2,3,5 and 6 had areas with high complexity for the machine, with lots of edges and noise (the sea in 2, a tree in 3, the low part of 5 and water in 6). These areas can be interpreted as of low visual complexity by humans, but for the metrics they are very complex areas with a lot of edges and changes in color.

In the case of the first outlier (see Fig. 9A) belonging to the AN group (abstract non-artistic), it may even seem reasonable to be detected as an outlier. This image, given its composition, as well as the nature of the metrics used, was interpreted by human as a repetitive texture or pattern, a kind of background, which may have been misleading to the system.

Surprisingly, the most initially promising technique, ENET, decreased its performance dramatically once the outliers were removed to an R-squared of 0.69. At this point, it was decided to review previous works using ENET and it was found that was initially developed to overcome the limitations of Lasso, and in general outperformed Lasso when the predictors were highly correlated (Zou & Hastie, 2005; Waldmann et al., 2013), as it had grouping effects and tended to select a group of highly correlated variables, once one variable was selected among them (Tibshirani, 1996; Bhlmann & Van de Geer, 2011). Furthermore, ENET tended to overfit the data in general, but more precisely when it was able only to build clusters of small sized co-variables. This occured because of the ENET attempts to explain the variation adding small noise variables to the clusters (Do, Quin & Vannucci, 2013), and this may seem plausible in our dataset, as there were 48 groups of correlated features with 7.8 ± 2.5 features each. Finally, the correlated variables were redundant in the sense that no additional information was obtained by adding them and sometimes they may insert noise in the clusters (Guyon et al., 2002).

However, SVM-RFE (0.69) reduced the whiskers although it seemed that a new outlier appeared in its boxplot (see Fig. 10) and FSMKL (0.71) was able to dramatically increase its performance without the outliers, reducing at the same time the variance between the experiments, but at the expense of the fact that some of them were no longer quartiles, as they became outliers. The other three methods also improved their results, or obtained similar ones, thus, it may seem that, given the lack of a set of unknown images to re-validate all models, the overfitting trend shown by ENET occurred with the images in our dataset.

It was found that the other three initially proposed methods (LM, GLM and Lasso) increase the R-squared performance if the 50 experiments were run again without the outliers and according to a pairwise Wilcoxon test (Wilcoxon, 1945), statistically significant with p-value < 8.08 × 10−9 for LM, p-value < 5.15 × 10−13 for GLM and p-value < 3.84 × 10−8 for Lasso. However, these results are, once again, worse than the ones obtained by ENET, FSMKL and SVM-RFE.

Given these results, the authors considered at this point that this strange behavior may only be related to the ENET. As can be seen in Fig. 10A, there is a significant difference (p-value < 2.2 × 10−16) between the results obtained by ENET and FSMKL with and without outliers according to a Wilcoxon test. The authors also checked the significance of the difference between ENET, FSMKL and SVM-RFE without outliers using a Friedman test with the Iman-Davenport extension. Our results showed that, with a very high level of confidence, FSMKL is significantly better than the other with a p-value < 1.92 × 10−41. Finally, in order to ensure the power and validity of our results, the contrast estimation is shown based on medians for the best three models in Fig. 10A using a heatmap. This estimation in non-parametric statistics is used to compute the real differences between the algorithms (García et al., 2010; Doksum, 1967).

Figure 10B shows the distribution of the residuals achieved in the correlation results of FSMKL. In order to clarify which particular group did each residual belong to, each dot was plotted in a different color. It should be noted that the slope of the AA, AM, RA and RN groups were very similar and almost parallel between them. However, the slope of the NHS group was steep and had a very different angle with respect to any other slope, in fact, this line intersected clearly all the others. This reveals the correlation between the first four groups and also the low or nonexistent correlation between those groups and NHS. The same was highlighted by Machado et al. (2015), where the authors stated that the content of the NHS group was largely cut off from the rest of the groups. In view of the FSMKL results, the authors of the current study agree with them.

Discussion

This study employed different ML approaches for visual complexity prediction. It was based on previous work by Forsythe et al. (2011) and we employed the metrics proposed in Machado et al. (2015). The main objective of this work is the study of other computational methods, in order to identify alternatives to ANN already used to solve this problem. So far, the best results obtained was 0.69 in terms of R-Squared using an input set of 329 features. From the studied methods, at least 3 offered similar results in terms of correlation coefficients using a significantly reduced number of metrics. FSMKL obtained 0.71 standing out with only 22 metrics, which were continuously repeated when performing 50 independent experiments and with an RMSE error with small variability. This stability of results allowed us to identify this method as the most reliable for the studied problem. FSMKL is an integrative kernel-based technique, since, in addition to identifying the metrics best adapted to the problem, it also looks for the relationships between them that best fit the objective of the experiment, making the most of the complementarity of the features to increase the obtained score. That is, it makes the most of the correlation between the variables to use them in a complementary way and to add more knowledge to the learning process. More specifically, it integrates information from different image descriptors and is able to weigh the degree of similarity of each subset, allowing in this way to select the features or feature sets of each of the groups that jointly obtain the best results.

As part of this study, the possible outliers of the most stable model were analyzed, according to which 5 images belonging to the NHS set and 1 image of the AN set were identified. These outliers may be due not only to their visual nature, but also to the descriptors used, since reliably detecting edges of both groups is a difficult task (Machado et al., 2015). It would be interesting to study other sets of descriptors not related to complexity estimators or edge detectors in order to check this issue. Taking into account the residual values of the resulting model, it was also possible to identify the difference between the NHS group and the rest. This difference is comprehensible since it consists of photographs, while the other four include paintings or cliparts, in which effects such as brushstroke are susceptible to identification by methods of border detection, whereas cliparts tend to be images less complex than a painting at computational level. Even the color channels can be different, with ’value’ more relevant in photographs than in cliparts or paintings.

Although it is not part of this study, it would be interesting to observe these computational models individually with each group (AA,AN,RA,RN,NHS). However, the few available examples may distort the results because of overtraining or overfitting problems, such as the one experienced with the ENET model in this paper, since the number of examples would be lower than the number of features. It would be necessary in this case to add an external cross-validation process, in order to avoid possible selection bias of the results during the process of selecting the best technique parameters (Ambroise & McLachlan, 2002).

This situation points out a limitation of the dataset employed. While it is the most interesting, it has some drawbacks. The total number of images is adequate for ML (taking into account the other options), but it includes very different types of images. Given the results of this work, one could note that the algorithms used for feature extraction depend on the type of images. In addition, paintings and designs (icons) are processed in the same way by computers and both types have a similar correlation with the human perception. But photographs tend to need a very different processing. The aim of further studies is to create two different datasets, one made up of photographs, and the other of the designs and paintings. Each dataset should include a high amount of images with different degree of complexity and aesthetic value. That would allow us to better understand the complexity measures that are more adequate for each type of image. Moreover, it would allow analyzing whether humans also employ different methods to calculate complexity.

Another approach may be simply to remove the NHS group, already identified as different, so that there would be 600 images divided into 4 groups, which may be compared pairwise according to different criteria.

By analyzing the twenty-two metrics that FSKML employ for complexity prediction, the authors suggest that edges, error of compression and Zipf’s Law are relevant for complexity estimation and can be related to the way human beings carry on the perception of visual complexity.

Previous studies, such as (Palmer, Schloss & Sammartino, 2013; Palmer & Schloss, 2010), have emphasized the color importance in many visual fields. The results presented in this paper show that certain channels of the HSV color model, in this case Saturation and Value, may be predominant in this type of problems. Both in this work and in Machado et al. (2015) it is clear that the ‘hue’ channel information is not necessary for computer complexity estimation, suggesting that the human being does not use this information for visual complexity.

In the same way, it is suggested that the canny edge detection filter is more informative than the sobel one, at least for complexity estimation since none of the sobel metrics has been employed by FSKML in the complexity prediction. The same applies to edge filters that only take into account one dimension (e.g., canny horizontal), which was not employed in any of the twenty-two features.

Machado et al. (2015) initially showed the validity of different features identified by three criteria: family or method, color channel and edge detection filters. The conducted experiments showed that, for example, in the case of JPEG- and Fractal-related features, all combinations of the S and V channels, with or without the Canny filter, provided the necessary information to obtain the best possible regression model. It was also found that the features related to the different methods for calculation of the Zipf’s Law complemented satisfactorily those already mentioned in the same conditions of color channels and edge detection filters.

The main limitations of this work include the use of a small set of ad-hoc metrics based on estimates from visual complexity performed by researchers. Further studies will explore other ad-hoc metrics and methods that will allow us to obtain metrics not created by humans, such as a layer from CNN (Carballal et al., 2019a). The difference between the maximum correlation (0.71) and the correlation between males and females (0.84), suggest the need for incorporating new different metrics.

Another interesting future line of work is to apply similar approaches to visual aesthetics. The first step would be to obtain a dataset of images, evaluated by a group of humans, taking into account their aesthetic value. For example, by asking a group of humans to evaluate the images of the Forsythe et al. (2011) dataset, it would be possible to (i) create a computer system that directly predicts the aesthetic value (ii) analyze the relationship between human-perceived visual complexity and aesthetics (iii) and analyze the relationship between predicted visual complexity and aesthetics.

Conclusions

This study proposed the application and validation of a set of different computational Machine Learning models for the evaluation of visual complexity. A set of 800 images were used (there were five different groups of images) and more than 300 features were calculated based on image compression error and Zipf’s Law over three color channels.

Six state-of-the-art Machine Learning regression algorithms were compared, with and without feature selection, and a final statistical analysis step was performed in order to evaluate the results and select the most promising one.

The novel FSMKL was chosen, and from an exhaustive outlier analysis, it was found that extreme images were present in the dataset and modified the regression value. Furthermore, it was proven that one of the groups containing 200 photographs of natural and human-made scenes (NHS) had a very low correlation with the other groups, as previously suggested by Machado et al. (2015).

Our results are of relevance, as they outperformed all the previous published works and are in accordance with the psychological findings of the human conception of visual complexity.

Supplemental Information

Dataset S1 and Dataset S2 DatasetS1 contains the training data used in the “RESULTS section”. DatasetS2 contains the training data used in the “OUTLIER REMOVAL section”

Click here for additional data file.

Additional Information and Declarations

Competing Interests

Author Contributions

Data Availability

The authors declare there are no competing interests.

Carlos Fernandez-Lozano and Adrian Carballal conceived and designed the experiments, performed the experiments, analyzed the data, prepared figures and/or tables, authored or reviewed drafts of the paper, approved the final draft.

Penousal Machado analyzed the data, contributed reagents/materials/analysis tools, approved the final draft.

Antonino Santos and Juan Romero conceived and designed the experiments, analyzed the data, contributed reagents/materials/analysis tools, approved the final draft.

The following information was supplied regarding data availability:

The raw data for this paper was provided by Marcos Nadal Roberts and Alex Forsythe, who have not given their permission to publish the data as part of this manuscript. This data was originally compiled for Marcos Nadal Roberts’s doctoral thesis, available at http://ibdigital.uib.cat/greenstone/collect/tesisUIB/index/assoc/TDX-0404.dir/TDX-0404108-112455.pdf. This data can be obtained from the corresponding author Alex Forsythe, email aof@aber.ac.uk (https://onlinelibrary.wiley.com/doi/abs/10.1348/000712610X498958) .

The scripts used to analyze Nadal and Forsythe’s data are our own and are available in Dataset S1 and Dataset S2.

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
