# Peer review of "Visual complexity modelling based on image features fusion of multiple kernels"

_PeerJ, doi:10.7717/peerj.7075_

## Round 0.1 · original submission · Major Revisions

Please improve the paper significantly by carefully addressing the issues raised by the reviewers.

Reviewer 1 ·

Basic reporting

In general the English is okay, but the writing could be improved at several places. For example, in l. 25, remove "the" in front of visual complexity; delete paragraph between l. 85/86; "influence of" is repeated in l. 144; change "have" to "has" in l. 385; l. 15, write number as 240; section number missing in l. 322; l. 156, it should say "5 seconds";

The abstract needs to be improved by i) mentioning the results of the study in much more details, ii) by rephrasing the first sentence which sounds too exaggerated, iii) by making it clear that this study is a re-analysis of data provided by Forsythe et al. 2011.

Throughout the manuscript there are incidences in which the names of authors are repeated in the text. For example, see l. 43, "Donderi Donderi (2006). The problem could be related to the reference software or the style template of PeerJ. In l. 33 the year is given once as 1932 and then as 1933. Please revise accordingly.

The beginning of the Discussion is too specific. I suggest that the authors repeat their main question and their main results in a few sentences before discussing details as color. The end of the Discussion would also profit from i) limitations of the current study, and ii) ideas for future research. Would it be possible to extend the Discussion and to discuss how the current results relate to perceptual/cognitive processes in more depth?

Research by Christopher Redies and Johan Wagemans on complexity could be added to the Introduction.

I could not open the figure files. In Figure 1 the year is missing after each reference, it seems. But this information would be useful to readers.

Experimental design

I suggest that the authors make it clear throughout the manuscript that this paper is based on a re-analysis of data collected in an earlier study. L. 124: This sentence reads as if the authors would have access to all data sets described in Figure 1, which is not the case. Please make it clear in the text that the authors' work - of course - also depended on the availability of data and that this also influenced the choice of the data set (besides the large N of stimuli etc.). Otherwise it is misleading.

Please do not mix information on the participants with information on the procedure (Section 0.3). L. 154, please provide more info on the definition of complexity (i.e., report it as it was given).

Figure 3: It is a bit unclear why the information about males/females is relevant for the current study. Please explain in the text.

L. 188: This sentences us unclear. What do the authors mean by "In order to design our experiments..."?

L. 205: I would not call the current study an experiment. Please explain why this term was used.

I like the rigorous approach to remove outliers. This section is important because it shows that we need to be careful because the results of the analysis may indeed change. I commend the authors on giving a detailed report.

Validity of the findings

Please provide more information in the Discussion on how the authors' results relate to human perception and cognition.

Reviewer 2 ·

Basic reporting

1.Try to set the problem discussed in this paper in more clear, write more to define the problem. So the reader can understand it deeply.
2.The discussion is insufficiency the author should be focus more on this section so that the reader can understand the importance of your work.
3.The paper's English needs to be significantly improved, as it its current form requires a lot of effort to read it.

Experimental design

The experimental design can be shown by formula rather than a simple language description.The content of the experiment can be rich and diverse, not like a report

Validity of the findings

There are no mentions about the complexity of the algorithm, as well as its comparison with other results from the literature.

Reviewer 3 ·

Basic reporting

There are some typo mistakes and I suggest authors to proof read the paper.
The paper is organized in a well manged format.

Experimental design

This authors proposed the application and validation of a set of different computational machine learning models for the evaluation of visual complexity. Six state-of-the-art Machine Learning regression algorithms were compared, with and without feature selection, and a final statistical analysis step was performed in order to evaluate the results and select the most promising one. The main contribution of this work bit limited and I my suggestion is incorporate the following changes before acceptance (My suggestion is minor revision)

Training and test time should be reported in the form of tables of figures.
The optimization details/parameters selection about machine learning algorithms are missing at the moment. It must be added at the time of revision.

Validity of the findings

A solid model is selected for evaluation of proposed research and main contribution of this work is bit limited (My suggestion is accepted marginally/minor revision) . My suggestions are to incorporate the following changes before formal acceptance.


1. There are some typo mistakes and I suggest authors to proof read the paper.
2. Training and test time should be reported in the form of tables of figures.
3. The optimization details/parameters selection about machine learning algorithms are missing at the moment. It must be added at the time of revision.
4. Some discussion about recent approaches based on deep-learning should be at least discussed in the paper and results should be compared even if they are less.
5. Some recent citations/research based on traditional machine learning approaches like SVM//ANN/Feature extraction should be discussed/cited at the time of revision.

“Intelligent Image Classification-Based on Spatial Weighted Histograms of Concentric Circles.”
“Image classification by addition of spatial information based on histograms of orthogonal vectors”
“A Hybrid Geometric Spatial Image Representation for scene classification”
“A novel image retrieval based on visual words integration of SIFT and SURF”
“A Novel Discriminating and Relative Global Spatial Image Representation with Applications in CBIR”
" Image retrieval by addition of spatial information based on histograms of triangular regions"

Additional comments

Dear Editor

Thanks a lot for sending me this manuscript for review. My suggestion is minor revision before formal acceptance.

Regards

Nouman Ali, PhD

---

## Round 0.2 · Minor Revisions

Please make the correction and provide response to the reviewers' comments.

Reviewer 1 ·

Basic reporting

l. 38, it should say "paperS"
l. 166, change "chosed" to "chose"
l. change to: access to "the" dataset used by Forsythe
l. 420, change to : work by Forsythe

There are several incidences in the Introduction where the in-text references are repeated, e.g., l.44, l.61, l.114, l.116, l.129... please check the complete manuscript.

Figure 9, E, typo in "outliner2 5

Experimental design

no comment

Validity of the findings

no comment

Additional comments

The authors have considerably improved their manuscript. I suggest that the authors read carefully through the manuscript to check for last language and formatting issues.

Reviewer 2 ·

Basic reporting

The author propose the application and validation of a set of different computational Machine Learning 498 models for the evaluation of visual complexity. I think the article is innovative.The application direction is also relatively new.The paper’s English has been improved.

Experimental design

The author has add some discussion about recent approaches based on machine learning have been discussed in the paper. SVM//ANN/Feature extraction have been added.The content of the experiment has been improved.

Validity of the findings

More info about models complexity and computing time has been added. The training and test time has be reported in the form of tables of figures.

Reviewer 3 ·

Basic reporting

English used in the paper is standard. Results presented are significant. However, i will suggest 02 minor changes.
1. The quality of figures should be improved , at present, the figures are in low quality.
2. Following important relevant citation based on feature extraction are missing, I strongly suggest authors to cite the following papers.

'Multi-pyramid image spatial structure based on coarse-to-fine pyramid and scale space'
"Image retrieval by addition of spatial information based on histograms of triangular regions"
'Fast feature matching based on r-nearest k-means searching'
“Intelligent Image Classification-Based on Spatial Weighted Histograms of
Concentric Circles.”
'Fast object detection based on binary deep convolution neural networks'
“A Novel Discriminating and Relative Global Spatial Image Representation with
Applications in CBIR”

Experimental design

The authors have significantly improved this section.

Validity of the findings

The authors have significantly improved this section.

Additional comments

The quality of figures may please be improved and it is suggested to cite and discuss the following relevant citations.

'Multi-pyramid image spatial structure based on coarse-to-fine pyramid and scale space'
'Fast feature matching based on r-nearest k-means searching'
'Fast object detection based on binary deep convolution neural networks'
“Intelligent Image Classification-Based on Spatial Weighted Histograms of
Concentric Circles.”
“A Novel Discriminating and Relative Global Spatial Image Representation with
Applications in CBIR”
"Image retrieval by addition of spatial information based on histograms of triangular regions"

---

## Round 0.3 · accepted · Accept

Please use the final review comment to fix that typo while in production.

Reviewer 1 ·

Basic reporting

no comment

Experimental design

no comment

Validity of the findings

no comment

Additional comments

I am happy with the latest version of the manuscript.

Reviewer 3 ·

Basic reporting

The authors have addressed my all suggestions and quality of this manuscript now very good. Therefore, I suggest the respected Editor to please accept this manuscript for publication.

Experimental design

Research question well defined, relevant & meaningful.

Validity of the findings

Data is robust, statistically sound, & controlled. Conclusions are well stated, linked to original research question & limited to supporting results.

Additional comments

I observed a typo error in the references section that I copied here as

'Wang, K., Zhu, N., Cheng, Y., Li, R., Zhou, T., and Long, X. (2018). Fast feature matching based on759 ¡i¿r¡/i¿-nearest ¡i¿k¡/i¿-means searching. CAAI Transactions on Intelligence Technology, 3:198–207(9).'

This is in the line 759-760 from references section. Respected Authors are suggested to please correct the above mentioned type error before sending the camera ready version for publication.